# Tributyrin in Inflammation: Does White Adipose Tissue Affect Colorectal Cancer?

**DOI:** 10.3390/nu11010110

**Published:** 2019-01-08

**Authors:** Luana Amorim Biondo, Alexandre Abilio S. Teixeira, Loreana S. Silveira, Camila O. Souza, Raquel G. F. Costa, Tiego A. Diniz, Francielle C. Mosele, José Cesar Rosa Neto

**Affiliations:** 1Department of Cell and Developmental Biology, Institute of Biomedical Sciences, University of São Paulo (USP), Av. Lineu Prestes, 1524-lab.435, 05508-000 São Paulo, SP, Brazil; alexandreast@gmail.com (A.A.S.T.); loreana_loly@hotmail.com (L.S.S.); camilaoliveira.biomed@gmail.com (C.O.S.); raquel.galfig@gmail.com (R.G.F.C.); tiegodiniz@gmail.com (T.A.D.); francmosele@gmail.com (F.C.M.); 2Department of Physical Education, Exercise and Immunometabolism Research Group, Post-Graduation Program in Movement Sciences, Universidade Estadual Paulista (UNESP), Rua Roberto Simonsen, 305, 19060-900 Presidente Prudente, SP, Brazil

**Keywords:** colon carcinogenesis, butyrate, white adipose tissue, fructooligosaccharides

## Abstract

Colorectal cancer affects the large intestine, leading to loss of white adipose tissue (WAT) and alterations in adipokine secretion. Lower incidence of colorectal cancer is associated with increased fibre intake. Fructooligosaccharides (FOS) are fibres that increase production of butyrate by the intestinal microbiota. Tributyrin, a prodrug of butyric acid, exerts beneficial anti-inflammatory effects on colorectal cancer. Our aim was to characterise the effects of diets rich in FOS and tributyrin within the context of a colon carcinogenesis model, and characterise possible support of tumorigenesis by WAT. C57/BL6 male mice were divided into four groups: a control group (CT) fed with chow diet and three colon carcinogenesis-induced groups fed either with chow diet (CA), tributyrin-supplemented diet (BUT), or with FOS-supplemented diet. Colon carcinogenesis decreased adipose mass in subcutaneous, epididymal, and retroperitoneal tissues, while also reducing serum glucose and leptin concentrations. However, it did not alter the concentrations of adiponectin, interleukin (IL)-6, IL-10, and tumour necrosis factor alpha (TNF)-α in WAT. Additionally, the supplements did not revert the colon cancer affected parameters. The BUT group exhibited even higher glucose tolerance and levels of IL-6, VEGF, and TNF-α in WAT. To conclude our study, FOS and butyrate supplements were not beneficial. In addition, butyrate worsened adipose tissue inflammation.

## 1. Introduction

Colorectal cancer is a common cancer associated with the occidental lifestyle, characterised by high caloric consumption and low physical activity [1]. Low consumption of fibre and high consumption of processed food are major risk factors for colorectal cancer [2,3]. Epidemiological data has shown an inverse association between total dietary fibre content and colon cancer risk [4], demonstrating that high fibre intake was associated with a lower risk of colon cancer [5]; however, the type of fibre that is ingested is also important [6].

In this sense, fructooligosaccharides (FOS) are linear short chains of fructose found in a wide range of natural foods, such as yacòn, garlic, bananas, wheat, etc. [7], that are unable to be digested by the upper gastrointestinal tract prior to reaching the colon [8]. These soluble dietary fibres are fermented by intestinal microbiota, producing short chain fatty acids (SCFAs): acetate, propionate, and butyrate. These SCFAs reduce luminal pH and can inhibit the growth and activity of pathogenic bacteria, potentially improving the host’s health [8].

Butyrate is an abundant metabolite found in the intestinal lumen, and it plays an essential role in the maintenance of gut homeostasis [9]. Butyrate is produced by two families of human colonic Firmicutes, Ruminococcaceae and Lachnospiraceae, from carbohydrates via glycolysis [10]. Butyrate is the main source of energy for enterocytes, allowing production of ATP, and exhibiting anti-inflammatory and anti-carcinogenic activity [11]. It can produce beneficial effects on colorectal cancer by inhibiting the growth and proliferation of tumour cells, in addition to its pro-apoptotic activities and local anti-inflammatory effects [11,12].

Tributyrin is an analogue of butyrate that can be safely administered by mouth, and it is composed of three molecules of butyrate esterified to a glycerol [13]. In vitro, tributyrin induces apoptosis by activating caspase-3 [14]. In addition, FOS is a safe supplement that helps elevate SCFA levels in the intestinal lumen [15].

In colonocytes, butyrate is an energetic precursor; however, in immune system cells butyrate plays a different role. Intestinal macrophages and dendritic cells reduce pro-inflammatory cytokines and promote secretion of anti-inflammatory agents like interleukin (IL)-10 and IL-18. These effects are mediated through inhibition of histone deacetylates (HDACs), altering the expression of genes necessary for apoptosis and cell proliferation and differentiation [11].

Another important risk factor for colon cancer is pre-diagnosis of inflammatory bowel disease (IBD) [16]. Approximately 20% of patients with IBD eventually develop colon cancer, and this population has a twofold increase in mortality [17,18].

In patients with cancer, loss of subcutaneous and visceral adipose tissue is associated with reductions in quality of life and survival [19]. Fat loss changes homeostasis, producing energy imbalances and modifying endocrine secretion and cytokine production [19,20]. White adipose tissue (WAT) is composed of adipocytes, stromal cells, vascular cells, endothelial cells, pericytes, and pluripotent cells, which are responsible for establishing the local microenvironment [21]. It is well known that the recruitment of macrophages and lymphocytes from the immune system increases the secretion of inflammatory factors; however, the role of adipocytes in tumorigenesis is a relatively new area of investigation [21].

Adipocytes are the main source of cytokines and adipokines for cancer cells [22]. Kannen et al. [23] verified that, after partial lipectomy, mice exhibited fewer pre-neoplastic lesions, suggesting that adipose tissue promotes tumorigenesis. In the context of obesity, low-grade inflammation and secretion of hypertrophic adipocytes produces dysfunction in mitochondrial colonocytes, directly affecting epithelial cell metabolism and cancer progression [22].

Therefore, nutritional strategies to prevent colorectal cancer development are needed. Our aim was to characterise the effects of FOS and tributyrin dietary supplementation within the context of a colon carcinogenesis model and determine the possible effects of WAT on tumorigenesis. The conclusion of our study is that FOS and butyrate supplements were not able to provide notable benefits and butyrate worsened adipose tissue inflammation. 

## 2. Materials and Methods

### 2.1. Animals

The Experimental Research Committee of the University of São Paulo approved all procedures relating to the care of animals used in this study. All experiments were performed in accordance with the approved guidelines of the Institute of Biomedical Sciences/University of São Paulo (ICB-USP) Animal Ethics Committee, registered under number 43/2016 of this Institute.

C57/BL6 mice, approximately 8 weeks of age (weighing 19–26 g), were obtained from the Biomedical Sciences Institute of the University of São Paulo and were housed two to three animals per cage in an animal room under a 12:12 h light–dark cycle.

### 2.2. Experimental Procedures

Animals were divided into four groups: a control group (CT) group fed a chow diet, and three experimental groups. The colon carcinogenesis group was fed a chow diet (CA), the fructooligosaccharides (FOS) group was fed a FOS-supplemented diet, and the tributyrin (BUT) group was fed a tributyrin-supplemented diet. Mice in the three experimental groups were fed their respective diets (chow diet, diet rich in FOS or BUT) for 12 weeks. All diets were AIN-93G-based. The compositions of the diets are listed in Table 1. 

### 2.3. Colon Carcinogenesis Induction (AOM/DSS)

After the 4-week diet intake period, the mice were administered intraperitoneal injections of 10 mg/kg body weight (BW) of azoxymethane (AOM) dissolved in physiological saline [24,25]. In addition, 2% dextran sodium sulphate (DSS) was administered via drinking water for 5 days, followed by 15 days of regular water [26]. DSS at a molecular weight of 40,000–50,000 Da and AOM were obtained from Sigma^®^ (Sigma Aldrich^®^, St. Louis, MO, USA). This cycle was repeated three times (Figure 1). 

BW, food intake, and water consumption were measured once a week. The animals’ health and well-being were monitored three times a week.

After the third cycle of AOM/DSS, the animals were euthanised by decapitation, and adipose tissue depots, livers, and gastrocnemius were removed, weighed, flash-frozen in liquid nitrogen, and stored at −80 °C. Blood was drawn and centrifuged, and serum was removed and kept frozen at −80 °C for further analysis.

### 2.4. Lipid and Glucose Levels

Serum levels of total cholesterol, triacylglycerol, and glycaemia were determined using enzymatic methods (Labtest^®^, Lagoa Santa, MG, Brazil). For the quantitative determination of non-esterified fatty acids in the serum, a colorimetric method assay was employed (NEFA-HR, Wako^®^, Mountain View, CA, USA).

### 2.5. Glucose Test Tolerance

After 11 weeks of diet treatment and colon carcinogenesis induction, the mice were fasted for 6 h prior to receiving intraperitoneal injections of glucose (2 g/kg BW). Blood samples were collected from the tail vein before and at 15, 30, 60, and 90 min post-glucose injection. The levels of glucose were measured by Accu-Chek^®^ Performa glucometer (Roche^®^, São Paulo, SP, Brazil). Differences in glycaemia before and during glucose administration were used to calculate the area under the curve (AUC).

### 2.6. Enzyme Linked Immunosorbent Assay (ELISA)

Quantitative assessments of adiponectin, leptin, interleukin 6 (IL-6), tumour necrosis factor alpha (TNF-α), interleukin 10 (IL-10), and vascular endothelial growth factor (VEGF) were performed by ELISA (DuoSet ELISA^®^, R&D Systems, Minneapolis, MN, USA).

### 2.7. Colon Histology

Small fragments of colon were fixed in paraformaldehyde (10%), embedded in paraffin, and serially cross-sectioned into 5-μm-thick sections. The sections were stained with haematoxylin and eosin (H&E) for morphological analyses. The 40× and 100× digital images were captured using an optical microscope with an attached AxioCamm HRC (Carl Zeiss^®^, São Paulo, SP, Brazil).

Crypt size quantification was performed using 40× digital images of five independent sections of each mouse. The images were representative of regions where there were no polyps or tumorous tissue.

Quantification of eosinophils, plasma cells, and fibroblasts in the lamina propria in colon sections subjected to H&E staining were performed. This analysis was based on cell-type morphology by light microscopy using the 100× digital images. The lamina propria digital images were representative of regions where there were no polyps or tumorous tissue.

### 2.8. Statistical Analysis

Statistical analyses were performed using the GraphPad Prism statistics software package version 5.0 for Windows (GraphPad Software, San Diego, CA, USA). The data are expressed as means ± SEM. Outlier data were detected by Grubbs’ test, removed, and then data were analysed using the ANOVA one-way test with Bonferroni post hoc testing. Values of *p* < 0.05 (a), *p* < 0.01 (b), *p* < 0.001 (c), and *p* < 0.0001 (d) were considered statistically significant.

## 3. Results

We measured BW differences between the control group and CA group after the first DSS cycle. However, the FOS showed decreased BW after the third cycle of DSS, whereas the BUT group showed reduced BW throughout the entire protocol (Figure 2A). After the carcinogenesis protocol, BW gains were reduced in the FOS group, compared to the control group, and in the BUT group, BW was decreased in relation to all of the other experimental groups (Figure 2B).

Food consumption was similar among the groups (Figure 2C). Mice who consumed the FOS-supplemented diet showed elevated water consumption (Figure 2F), and tributyrin diet supplementation was associated with reduced survival (Figure 2D). Leptin was reduced in all groups with colon carcinogenesis (Figure 2E).

Colon carcinogenesis reduces the retroperitoneal, epididymal, and brown adipose depots. The butyrate diet markedly reduced the adipose tissues mass (Figure 3A–E), gastrocnemius skeletal muscle mass, and liver (Figure 3F,G) weight. In contrast, total cholesterol, triacylglycerol, and non-esterified fatty acid (NEFA) in serum were unchanged (Figure 4A–C).

During glucose intolerance analysis, the tributyrin-supplemented group showed reduced glycaemia at the 0, 15, 60, and 90 min time points. These results suggest that butyrate promoted glucose tolerance (Figure 5A). However, the colon carcinogenesis protocol, and the different diets, did not alter glucose tolerance (Figure 5B). Basal glycaemia was reduced in all the cancer groups, without apparent treatment effects (Figure 5C).

Serum adiponectin was similar across the various groups (Figure 6E); however, adipokine was elevated in subcutaneous and retroperitoneal adipose tissues in mice within the BUT group (Figure 6A,B).

In our study, 100% of the mice from the cancer groups demonstrated similar macroscopically polypoid tumours. Figure 7 depicts the difference between the epithelium taken from the control and cancer groups. Histologically, the cancer groups showed that abnormal colon epithelial phenotypes, polypoid tumours on the mucosal surface, elevated lymphoid aggregates, and loss of polarity and butyrate led to colonic crypt hypertrophy.

Analysis of the plasma cell infiltrate in the intestinal lamina propria showed a lower percentage of plasma cells in the colon of the animals in both the CA and FOS groups (Figure 8C), compared with the control mice. Interestingly, butyrate supplementation increased plasma cell distribution in the animals’ colons, like that observed in the control group.

We observed more fibroblasts in the colons of the animals supplemented with butyrate, compared to the FOS group (Figure 8D). There were no significant between-group differences in eosinophil infiltration within the colon (Figure 8B).

While AOM is genotoxic to colonic epithelial cells and DSS-induced colitis, induction of colon carcinogenesis did not promote pro-inflammatory markers; however, reduced IL-10 content was observed in the colon (Figure 9A–D).

Interestingly, in mice that received a tributyrin-supplemented diet, we observed elevation of IL-6 and TNF-α, principally in retroperitoneal and epididymal adipose tissue (Figure 10D,E,G,H). In subcutaneous tissue, IL-6 alone was elevated (Figure 10A). IL-10 was elevated only in retroperitoneal adipose tissue within mice who consumed a tributyrin-supplemented diet (Figure 10F). Meanwhile, mesenteric adipose tissue was not affected by induction of colon carcinogenesis (Figure 10J–L).

Tributyrin supplementation induced elevation of VEGF in retroperitoneal adipose tissue and reduced VEGF in mesenteric depot (Figure 11A–D).

## 4. Discussion

Induction of colon carcinogenesis reduced BW gains and loss of WAT mass. Under microscopy, colon tissue presented with dysplastic lesions, adenomas, tumours, and crypt hypertrophy. Diet treatments did not exert a protective effect, and tributyrin diet supplementation was associated with reduced survival and inflammation within WAT.

Colon cancer reduced BW by ~12% and butyrate supplementation reduced weight gain by ~38%, compared with the control group. FOS and BUT failed to prevent adipose tissue loss and low leptin levels. Similar to our cancer group, in patients with cancer, weight loss and reduced leptin levels are common [27].

Food intake did not increase as expected, probably because ghrelin, another hormone with orexigenic properties, is decreased in patients with cancer [27]. Moreover, complex pathways regulate food intake. Thus, the inhibition of fatty acid oxidation in the hypothalamus reduces fatty acid synthases and consequently promotes elevation of malonyl-coenzyme A content, effectively blocking orexigenic peptides [28]. Besides its neuroendocrine actions, leptin stimulated growth and proliferation in the colorectal carcinogenesis model [29].

We observed disruptions on adipose mass in retroperitoneal and epididymal pads induced by colon cancer, and in the subcutaneous pad caused by tributyrin treatment. The reductions on adipose mass resulted in alterations on the secretion of leptin, adiponectin, and cytokines, leading to disruption of glucose, lipid metabolism, insulin signalling, and inflammation [30,31]. In a chronic illness, such as cancer-related cachexia, the catabolism of proteins and mobilisation of lipids from adipose tissue compromises organ and tissue functions, worsening prognosis and quality of life [27].

Reduced glycaemia occurs because cancerous colonocytes prefer glucose metabolism as an energy source. This increases the uptake of glucose by elevating expression of glucose transporters [32]. GLUT-1 is an indicator of disease aggression and metastasis in patients with colorectal cancer [33]. In our study, neither supplement was able to restore fasting blood glucose levels induced by metabolic changes during colorectal cancer.

Colorectal cancer did not change glucose homeostasis, and butyrate reduced the AUC of the glucose tolerance test. Aguilar et al. [34] found that oral glucose tolerance was improved in lean mice treated with sodium butyrate. In mice with diet-induced obesity, Gao et al. [35] found that butyrate improved insulin resistance by increasing mitochondrial biogenesis in brown adipose tissue by enhancing uncoupling protein (UCP)-1 and peroxisome proliferator–activated receptor (PPAR)-γ coactivator (PGC)-1α protein content, and enhancing type 1 fibres. Increased glucose tolerance, induced by butyrate, could be explained by the elevation of adiponectin content in subcutaneous and retroperitoneal adipose tissues. Adiponectin is a hormone insulin sensitiser; through APPL1, adiponectin activates AMPK and MAPK38, leading to glucose uptake and improving glucose homeostasis [36].

There is no consensus as to the role of adiponectin in experimental colon cancer models. Ealey and Archer [37] showed adiponectin transgenic mice with elevated circulating adiponectin were not protected against the development of colon cancer induced by AOM. Boddicker et al. [25] and Saxena et al. [38] verified that adiponectin knockout mice exhibited more total lesions, when compared to wide type, in colorectal carcinogenesis AOM/DSS and DMH (1,2-dimethylhydrazine)/DSS experimental models, respectively.

Colorectal carcinogenesis is a multistep process with histological features and gene mutations that resemble human sporadic colon carcinogenesis [39]. In our study, the colon cancer groups exhibited macroscopically polypoid tumours and elevation of lymphoid aggregates. Butyrate supplementation led to crypt hypertrophy. Oral ingestion of butyrate can cause hypertrophy within small intestine villi [40].

Kim et al. [41] demonstrated that SCFAs, produced by the intestinal microbiota from the fermentation of dietary fibre, support host antibody responses by regulating the gene expression required for plasma B cell differentiation.

Butyrate is an energy source for enterocytes that prefer to metabolise fatty acids through β-oxidation in the mitochondria, avoiding entering the nucleus. In contrast, tumour cells prefer the metabolisation of glucose [32,42]. We utilised high doses of tributyrin, thus suggesting that β-oxidation was not an efficient process for the utilisation of butyrate, allowing the entrance of butyrate into the nucleus of cancerous and healthy enterocytes. In the nucleus, butyrate induces histone acetylation, modulating the conformation of nucleosomes and transcription of cytokine genes like NF-κB pathways [32]. Moreover, at this dose, butyrate leads to citrate formation, which can exit the mitochondria and be utilised for lipid synthesis in the cytosol [32].

Kaiko et al. [9] showed that colonocytes break down butyrate using oxidative phosphorylation through the expression of lipid metabolism enzymes, thereby reducing the contact between butyrate and intestinal stem cells and protecting the colon epithelial barrier. However, higher concentrations of butyrate in the base of the crypt supports cell proliferation and increases histone acetylation [32]. In the presence of neoplastic lesions, reduced mucus production and physiological protective factors like tight junctions may change the access of butyrate to the cells [43]. This facilitates use of butyrate as fuel, leading to epithelial hypertrophy.

It is interesting that the butyrate-supplemented group had consumed additional calorie, about 15 kcal/day, by tributyrin added in diet. However, this group showed severe reduction of body weight and adipose tissue mass. The literature showed butyrate is absorbed by rectal colon cells in special by the monocarboxylate transporters (MCT)-1 [44]. The butyrate absorption could be inhibited in tumour colonic cells, whereas MCTs are associated with the efflux and influx of lactate, which is increased in colon cancer cells [45]. Moreover, patients with intestinal bowel diseases display downregulation of bile acid uptake transporter [46], increased intestinal permeability, and disruption of intestinal barrier [47], demonstrating that bowel inflammation can alter digestion and nutrients uptake.

In contrast to our expectations, no differences were observed in regard to the inflammatory cytokines in the colons of the mice. Moreover, IL-10 was reduced in the tumour and butyrate groups. IL-10 is a regulatory cytokine with anti-inflammatory properties [48] and is secreted in large amounts by T regulatory lymphocytes. This cytokine helped sustain M2 polarisation [49]. Increased numbers of IL-10 and M2 macrophages increase the proliferation and malignancy of colon tumours [50] by inhibiting the activity of natural killer cells and dendritic cells. In our experimental model, IL-10 was decreased in colon tumours. This finding can be explained because our model resembles the progression of gut inflammatory disease with tumour formation as the end of this progressive disease. In this case, IL-10 was able to reduce that inflammation and was associated with a better prognosis than when the dendritic cells were pretreated with IL-10 [51].

Our results additionally showed that this protocol led to cancer cachexia, a syndrome characterised by weight loss, anorexia, disruption of adipose tissue, and reduction of skeletal muscle mass, including pro-inflammatory cytokines and hypermetabolism [52]. Dietary supplementation with 5% tributyrin for 10 months was not toxic [53]; however, 10% could be toxic to adipose tissue. Butyrate can damage lipid metabolism and influence inflammation. In our study, butyrate elevated IL-6 and TNF-α in white adipose depots.

Tributyrin supplementation additionally elevated VEGF in retroperitoneal tissues and reduced it in mesenteric adipose tissue. The STAT3–VEGF pathway is essential for colon cancer development; adiponectin can reduce mRNA VEGF and signal transducer and activator transcription 3 phosphorylation (STAT3) [54]. Adiponectin administration suppresses implanted tumour growth in mice and can regulate the cell cycle, including proliferation, invasion, and inflammatory pathways in vitro [54]. Leptin can also activate the EGF/PI3K/STAT3 (epidermal growth factor – EGF, phosphoinositide 3-kinase – PI3, signal transducer and activator transcription 3 – STAT3) signalling pathways necessary for neoplastic progression. EGF induces STAT3 binding within VEGF and leptin promoters, stimulating leptin and VEGF gene and protein synthesis [55]. STAT3 is also involved in the secretion of IL-6 and other inflammatory markers [56]. In our study, the STAT3 pathway was activated in the butyrate group, as IL-6, TNF-α, and VEGF were higher expressed.

This inflammation of adipose tissue is common in cachexia syndrome, resulting in a deep reduction of adipose tissue mass in association with elevation of inflammatory cytokines. Tsoli et al. [57] verified higher gene expression of STAT3 and SOCS3 in the WAT of colon-26 tumour-bearing mice that also presented with higher concentrations of circulating IL-6. Furthermore, adipokines and cytokines secreted from adipose tissue (together with lipids from adipocytes present in tumour environment) support tumour progression [58].

In patients with cancer, there is strong evidence of a linkage between fat mass and colon cancer. However, the molecular mechanisms that govern this linkage are not fully understood. Adipose tissue is probably a trigger and sustains tumorigenesis by producing pro-inflammatory cytokines, growth factors, metalloproteinases, and proangiogenic factors, as demonstrated by our findings of increased VEGF [1,59]. Together, these factors induce a higher risk of colon cancer in humans [1]. Recent evidence suggests that IL-32α mRNA and protein levels are upregulated in the adipose tissue of patients with colon cancer. Circulating levels are also increased. The coculture between the adipose tissue of obese patients and immortalised colon cancer humans cells showed increased expression of IL-32α in medium and greater proliferation of tumour cells [60].

However, this association between adipose tissue inflammation and tumorigenesis is found in other types of cancer. WAT inflammation is associated with increased tumour thickness and vascular invasion in patients with early stage oral tongue carcinoma [61].

In conclusion, the inflammation of WAT and adipokines was not modulated by colon carcinogenesis. Dietary supplementation of FOS and tributyrin did not have a beneficial effect on colon carcinogenesis. Moreover, butyrate worsened adipose tissue inflammation, and the group that received butyrate supplementation showed increased crypt hypertrophy and total neoplastic area. It was possible to establish a link between adipose tissue inflammation and more aggressive tumour growth in the butyrate group.

## Figures and Tables

**Figure 1 nutrients-11-00110-f001:**
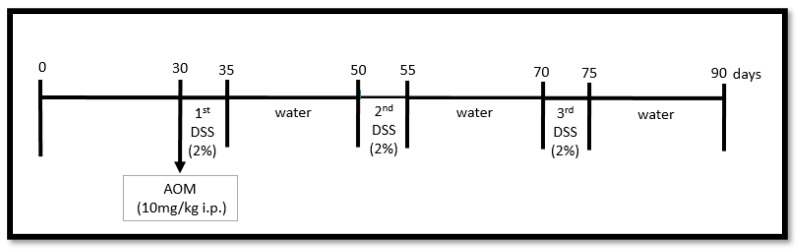
Colon carcinogenesis experimental protocol: colon carcinogenesis was induced chemically by a single dose of azoxymethane (AOM), with 10 mg/kg body weight administered intraperitoneally on the 30th day, and subsequently three cycles of dextran sodium sulphate (DSS) consisted of 5 days drinking DSS and 15 days drinking water. The diet started on day 0 and finished on the 90th day.

**Figure 2 nutrients-11-00110-f002:**
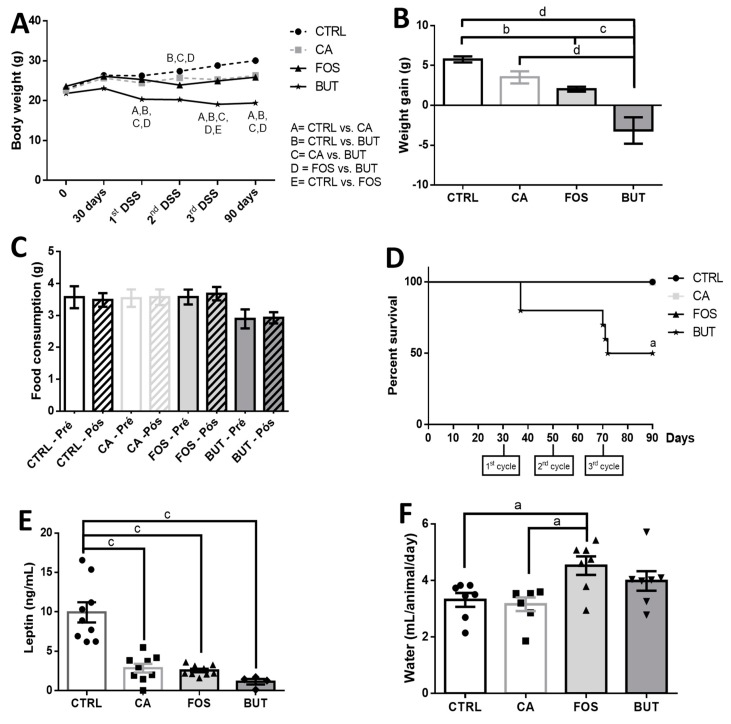
Butyrate reduces body weight and leptin: evolution of body weight (**A**), weight gain (**B**), food consumption (**C**), percent survival (**D**), serum leptin (**E**), and water intake (**F**). Mice were divided into four groups: control group (CTRL—*n* = 9–10) fed with chow diet; and colon carcinogenesis-induced groups fed either with chow diet (CA—*n* = 9–10), tributyrin-supplemented diet (BUT—*n* = 5–6), or with FOS-supplemented diet (FOS—*n* = 9–10). Colon carcinogenesis was induced chemically by azoxymethane (AOM)/dextran sodium sulphate (DSS). Data are presented as means ± SEM. The groups were compared using one-way ANOVA followed by Bonferroni post hoc tests; for *p* < 0.05 (a), *p* < 0.01 (b), *p* < 0.001 (c), and *p* < 0.0001 (d); black circles (●), square (∎), triangles (▲), inverted triangles (▼) represents sample numbers in the corresponding group.

**Figure 3 nutrients-11-00110-f003:**
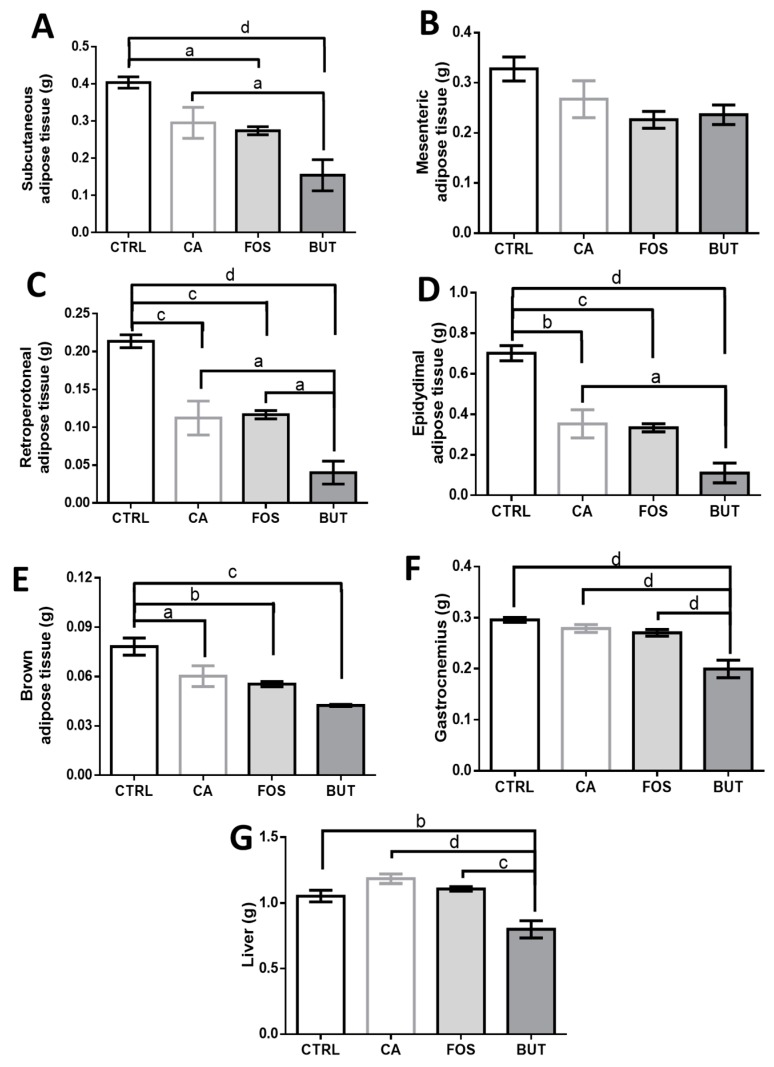
Butyrate reduces adipose tissue, gastrocnemius tissue mass, and liver weight: weights of subcutaneous (**A**), retroperitoneal (**B**), epididymal (**C**), mesenteric (**D**), brown (**E**), gastrocnemius skeletal muscle (**F**), and liver (**G**) tissue are presented below. Mice were divided into four groups: control group (CTRL—*n* = 9–10) fed with chow diet; and colon carcinogenesis-induced groups fed either with chow diet (CA—*n* = 9–10), tributyrin-supplemented diet (BUT—*n* = 5–6), or with FOS-supplemented diet (FOS—*n* = 9–10). Colon carcinogenesis was induced chemically by azoxymethane (AOM)/dextran sodium sulphate (DSS). Data are presented as means ± SEM. The groups were compared using one-way ANOVA followed by Bonferroni post hoc tests; for *p* < 0.05 (a), *p* < 0.01 (b), *p* < 0.001 (c), and *p* < 0.0001 (d).

**Figure 4 nutrients-11-00110-f004:**
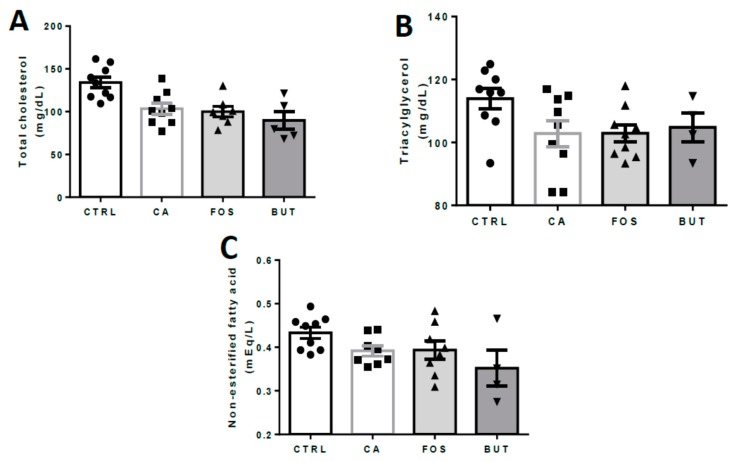
Lipid profile: total cholesterol (**A**), triacylglycerol (**B**), and non-esterified fatty acid (**C**). Mice were divided into four groups: control group (CTRL—*n* = 9–10) fed with chow diet; and colon carcinogenesis-induced groups fed either with chow diet (CA—*n* = 9–10), TB-supplemented diet (BUT—*n* = 5–6), or with FOS-supplemented diet (FOS—*n* = 9–10). Colon carcinogenesis was induced chemically by azoxymethane (AOM)/dextran sodium sulphate (DSS). Data are presented as means ± SEM. After outlier exclusion, the groups were compared using one-way ANOVA followed by Bonferroni post hoc tests; black circles (●), square (∎), triangles (▲), inverted triangles (▼) represents sample numbers in the corresponding group.

**Figure 5 nutrients-11-00110-f005:**
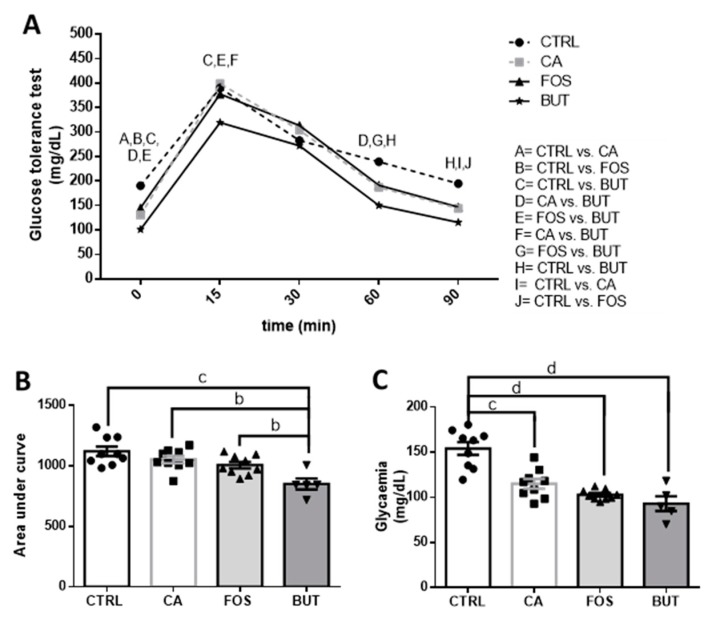
Butyrate reduces glycemia: glucose tolerance test (**A**), area under the curve (**B**), and glycaemia (**C**). Mice were divided into four groups: control group (CTRL—*n* = 9–10) fed with chow diet; and colon carcinogenesis-induced groups fed either with chow diet (CA—*n* = 9–10), TB-supplemented diet (BUT—*n* = 5–6), or with FOS-supplemented diet (FOS—*n* = 9–10). Colon carcinogenesis was induced chemically by azoxymethane (AOM)/dextran sodium sulphate (DSS). Data are presented as means ± SEM. The groups were compared using one-way ANOVA followed by Bonferroni post hoc tests; for *p* < 0.01 (b), *p* < 0.001 (c), and *p* < 0.0001 (d); black circles (●), square (∎), triangles (▲), inverted triangles (▼) represents sample numbers in the corresponding group.

**Figure 6 nutrients-11-00110-f006:**
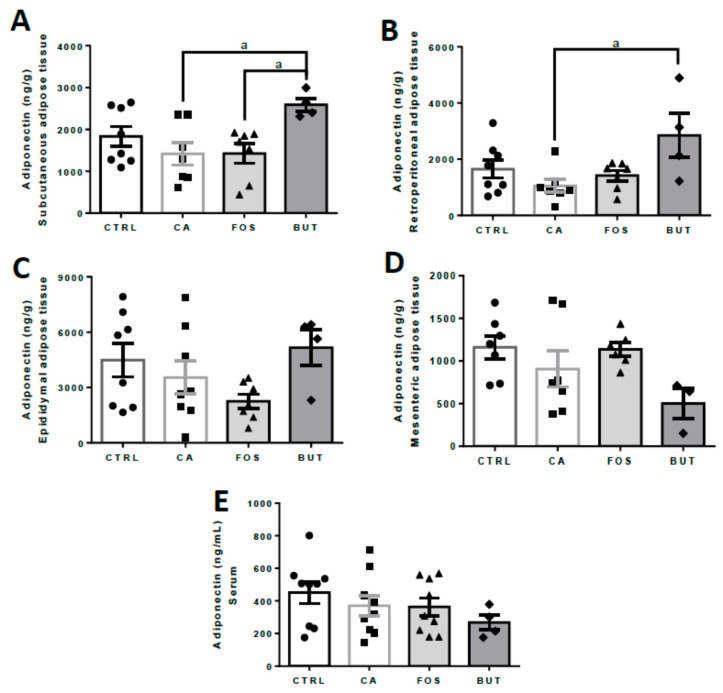
Butyrate elevates adiponectin content in subcutaneous and retroperitoneal adipose tissue: subcutaneous (**A**), retroperitoneal (**B**), epididymal (**C**), and mesenteric (**D**) adipose tissues, and serum (**E**). Mice were divided into four groups: control group (CTRL—*n* = 7–8) fed with chow diet; and colon carcinogenesis-induced groups fed either with chow diet (CA—*n* = 7–*8*), TB-supplemented diet (BUT—*n* = 4–5), or with FOS-supplemented diet (FOS—*n* = 7–8). Colon carcinogenesis was induced chemically by azoxymethane (AOM)/dextran sodium sulphate (DSS). Data are presented as means ± SEM. After outlier exclusion, the groups were compared using one-way ANOVA followed by Bonferroni post hoc tests; for *p* < 0.05 (a); black circles (●), square (∎), triangles (▲), rhombus (◆) represents sample numbers in the corresponding group.

**Figure 7 nutrients-11-00110-f007:**
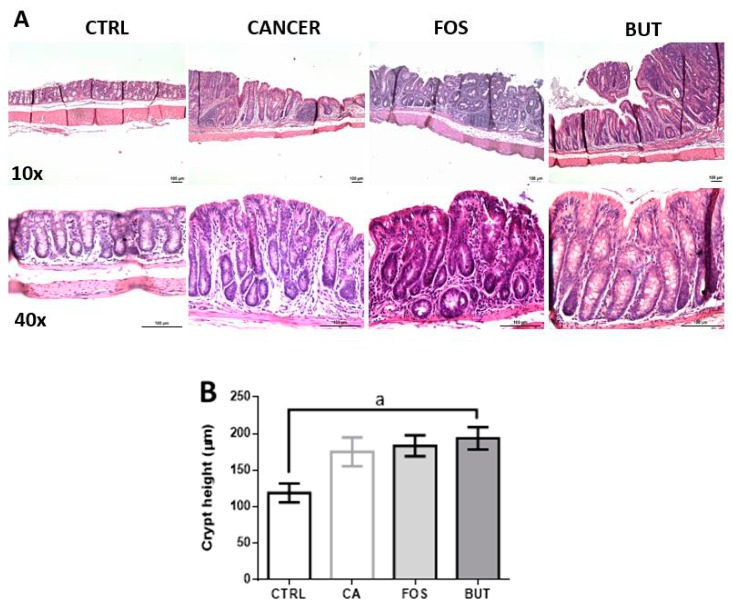
Butyrate promote hypertrophy of colonic crypts: Histological images of the colon (**A**) and colon crypt height (**B**). The tissues were subjected to haematoxylin and eosin (H&E) staining; the images represent 10× and 40× magnification; magnification bar: 100 µm. Mice were divided into four groups: control group (CTRL—*n* = 5) fed with chow diet; and colon carcinogenesis-induced groups fed either with chow diet (CA—*n* = 5), TB-supplemented diet (BUT—*n* = 5), or with FOS-supplemented diet (FOS—*n* = 5). Colon carcinogenesis was induced chemically by azoxymethane (AOM)/dextran sodium sulphate (DSS). Data are presented as means ± SEM. The groups were compared using one-way ANOVA followed by Bonferroni post hoc tests; for *p* < 0.05 (a).

**Figure 8 nutrients-11-00110-f008:**
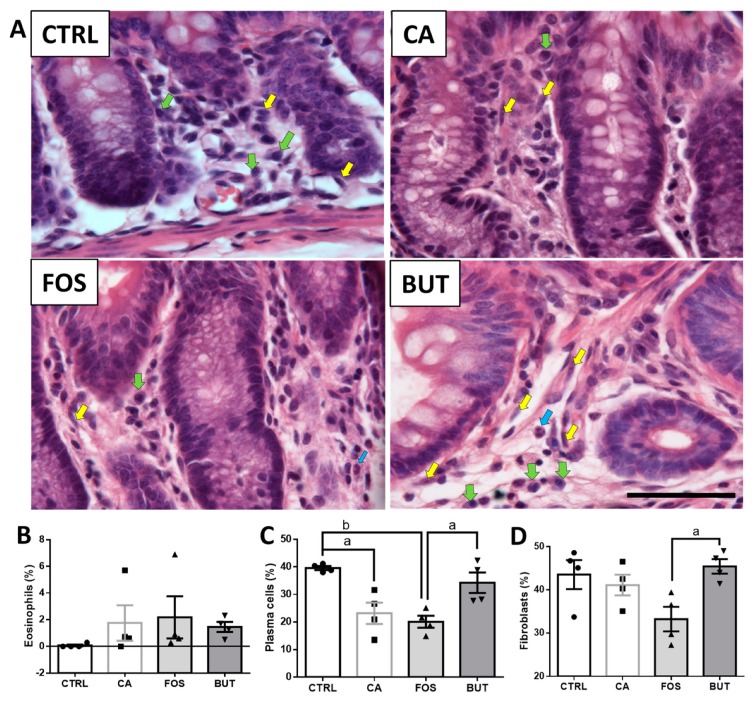
Cell infiltration in the lamina propria of the colon mucosa: Histological images of the colon (**A**) percentage of eosinophils (blue arrows, (**B**)), plasma cells (green arrows, (**C**)), and fibroblasts (yellow arrows, (**D**)) in all lamina propria cells. The tissues were subjected to H&E staining; the images represent 100 × magnification; magnification bar: 50 µm. Mice were divided into four groups: control group (CTRL—*n* = 5) fed with chow diet; and colon carcinogenesis-induced groups fed either with chow diet (CA—*n* = 5), TB-supplemented diet (BUT—*n* = 5), or with FOS-supplemented diet (FOS—*n* = 5). Colon carcinogenesis was induced chemically by azoxymethane (AOM)/dextran sodium sulphate (DSS). Data are presented as means ± SEM. The groups were compared using one-way ANOVA followed by Bonferroni post hoc tests; for *p* < 0.05 (a), *p* < 0.01 (b); black circles (●), square (∎), triangles (▲), inverted triangles (▼) represents sample numbers in the corresponding group.

**Figure 9 nutrients-11-00110-f009:**
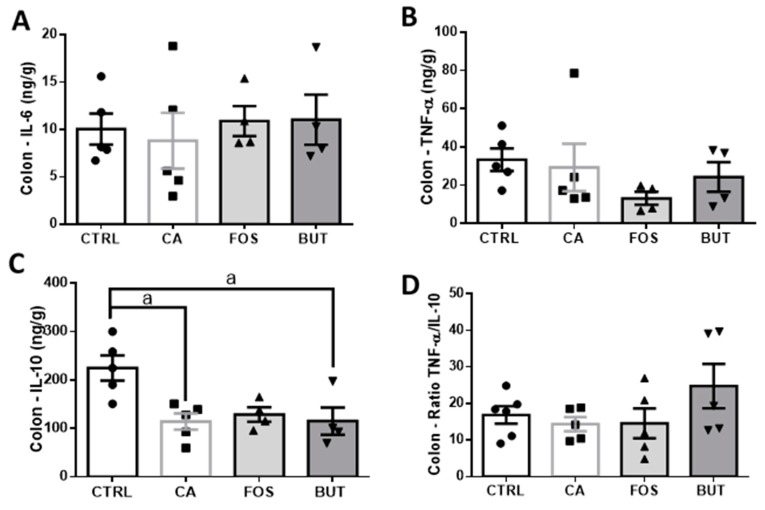
Colon carcinogenesis induction did not promote pro-inflammatory markers, but reduced IL-10 content in the colon: IL-6 (**A**), TNF-α (**B**), IL-10 (**C**), TNF-α/IL-10 ratio (**D**) in colon. Mice were divided into four groups: control group (CTRL—*n* = 5) fed with chow diet; and colon carcinogenesis-induced groups fed either with chow diet (CA—*n* = 5), TB-supplemented diet (BUT—*n* = 4), or with FOS-supplemented diet (FOS—*n* = 4). Colon carcinogenesis was induced chemically by azoxymethane (AOM)/dextran sodium sulphate (DSS). Data are presented as means ± SEM. The groups were compared using one-way ANOVA followed by Bonferroni post hoc tests; for *p* < 0.05 (a); black circles (●), square (∎), triangles (▲), inverted triangles (▼) represents sample numbers in the corresponding group.

**Figure 10 nutrients-11-00110-f010:**
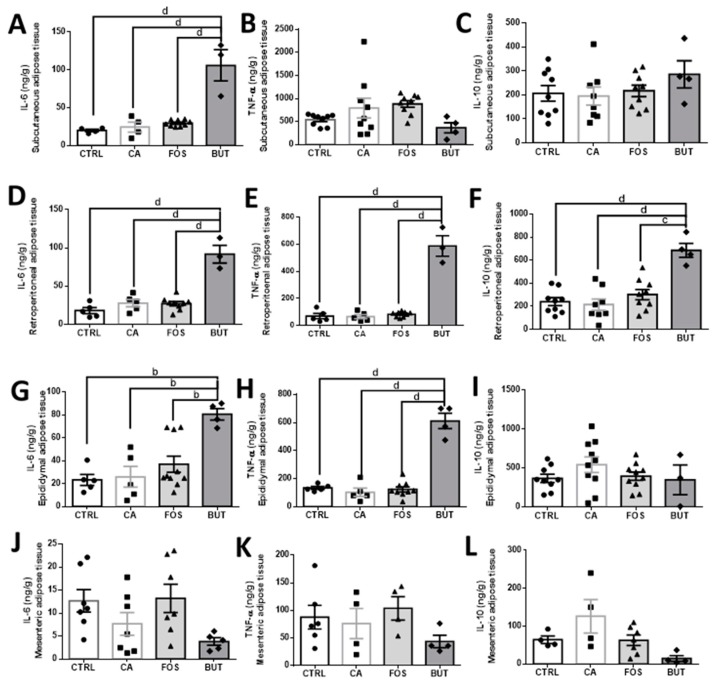
Butyrate promoted pro-inflammatory markers (IL-6 and TNF-α) and IL-10 in different adipose depots: IL-6, TNF-α, and IL-10 concentrations in subcutaneous (**A**–**C**), retroperitoneal (**D**–**F**), epididymal (**G**–**I**), and mesenteric (**J**–**L**) adipose tissues. Mice were divided into four groups: control group (CTRL—*n* = 6–9) fed with chow diet; and colon carcinogenesis-induced groups fed either with chow diet (CA—*n* = 4–9), TB-supplemented diet (BUT—*n* = 3–4), or with FOS-supplemented diet (FOS—*n* = 5–9). Colon carcinogenesis was induced chemically by azoxymethane (AOM)/dextran sodium sulphate (DSS). Data are presented as means ± SEM. After outlier exclusion, the groups were compared using one-way ANOVA followed by Bonferroni post hoc tests; for *p* < 0.01 (b), *p* < 0.001 (c), and *p* < 0.0001 (d); black circles (●), square (∎), triangles (▲), rhombus (◆) represents sample numbers in the corresponding group.

**Figure 11 nutrients-11-00110-f011:**
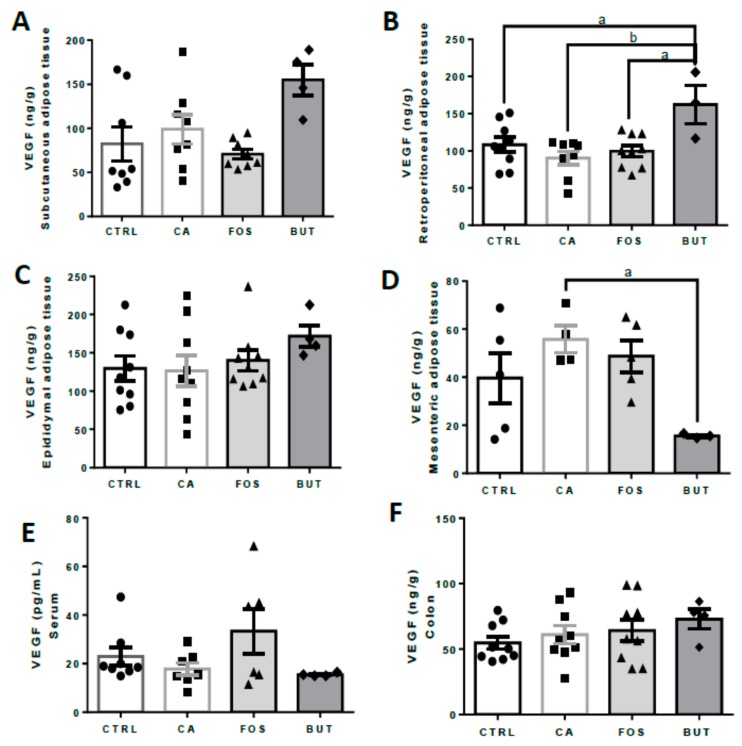
Butyrate promoted elevation of VEGF in retroperitoneal adipose tissue: VEGF concentrations in subcutaneous (**A**), retroperitoneal (**B**), epididymal (**C**), and mesenteric (**D**) adipose tissue, serum (**E**), and the colon (**F**). Mice were divided into four groups: control group (CTRL—*n* = 6–9) fed with chow diet; and colon carcinogenesis-induced groups fed either with chow diet (CA—*n* = 4–9), TB-supplemented diet (BUT—*n* = 3–4), or with FOS-supplemented diet (FOS—*n* = 5–9). Colon carcinogenesis was induced chemically by azoxymethane (AOM)/dextran sodium sulphate (DSS). Data are presented as means ± SEM. After outlier exclusion, the groups were compared using one-way ANOVA followed by Bonferroni post hoc tests; for *p* < 0.05 (a), *p* < 0.01 (b); black circles (●), square (∎), triangles (▲), rhombus (◆) represents sample numbers in the corresponding group.

**Table 1 nutrients-11-00110-t001:** Diet composition.

Products	Chow Diet (g/Kg)	Tributyrin Diet (g/Kg)	Fructooligosaccharides Diet (g/Kg)
Cornstarch	465	465	465
Dextrinized cornstarch	155	155	155
Casein	140	140	140
Sucrose	100	100	100
Soybean oil	40	40	40
Cellulose	50	50	50
Minerals mix	35	35	35
Vitamins mix	10	10	10
Choline bitartrate	2.5	2.5	2.5
L-cystine	1.8	1.8	1.8
Tributyrin	0	100	0
Fructooligosaccharides	0	0	60
Total energy (Kcal/Kg)	3474	4074	3474

Tributyrin—6 Kcal/g; FOS (Fructooligosaccharides)—0 Kcal/g.

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
