# Peer review of "Tributyrin in Inflammation: Does White Adipose Tissue Affect Colorectal Cancer?"

_nutrients, 2019, doi:10.3390/nu11010110_

Round 1
Reviewer 1 Report
Luana A. Biondo et al investigated if fiber or butyrate could regulate adipose tissue in a colorectal cancer mouse model. Interestingly, the authors observed that butyrate worsened the adipose tissue inflammation and fiber didn't lead to any significant benefit. The authors performed several experiments and observed differential changes. However, the authors need to have more mechanistic studies.
what is the rationale for the dosage of FOS and BUT? Low fiber intake has been shown to contribute to the increased risk of colorectal cancer. However, the relationship between high fiber intake and colorectal cancer is still controversial. The authors need to discuss more about this topic.
The authors need to discuss more about adipose tissue inflammation in colorectal cancer patients. What is the current research about this?
The authors observed differential changes in cytokines. They need to link the observation to adipose tissue inflammation in colorectal cancer, or in other types of cancer. What are the findings from cancer patients?
Author Response
About the dosage of FOS and BUT: The dosage of FOS was based in Perrin et al (2001) study, they showed that 6% of FOS in the diet reduced aberrant crypt foci in rats injected with azoxymethane and augmented concentration of butyrate in large intestine. The dosage of tributyrin was based in different studies, most of studies they use butyrate oral gavage in few weeks (three times/week), but in our study the treatment is longer than these finds (12 weeks) and we prefer the use of tributyrate in the diet. Deschner et al (1990) used tributyrin (5%) during 10 months and they not showed any toxicity, and how our treatment was in reduced period comparing to Deschner et al (1990), we preferred to use higher doses. Leonel et al (2013) found that low doses (0,5% of the diet) was not efficient to reduce IL-1β in colon mice.
· Deschner, E.E.; Ruperto, J.F.; Lupton, J.R.; Newmark, H.L. Dietary butyrate (tributyrin) does not enhance AOM-induced colon tumorigenesis. Cancer Lett. 1990, 52, 79–82.
· Perrin P, Pierre F, Patry Y, et al. Only fibres promoting a stable butyrate producing colonic ecosytem decrease the rate of aberrant crypti foci in rats. Gut. 2001; 48: 53-61.
· Leonel AJ, Teixeira LG, Oliveira RP, Santiago AF, Batista NV, Ferreira TR, Santos RC, Cardoso VN, Cara DC, Faria AM, Alvarez-Leite J. Antioxidative and immunomodulatory effects of tributyrin supplementation on experimental colitis. Br J Nutr. 2013 Apr 28;109(8):1396-407.
What is the current research about this adipose tissue inflammation in colorectal cancer patients?
It is a good question and we inserted a paragraph about this topic.
This inflammation of adipose tissue is common in cachexia syndrome, resulting in a deep reduction of adipose tissue mass in association with elevation of inflammatory cytokines. Tsoli et al. [53] verified higher gene expression of STAT3 and SOCS3 in WAT of colon-26 tumour-bearing mice that also presented with higher concentrations of circulating IL-6. Furthermore, adipokines and cytokines secreted from adipose tissue (together with lipids from adipocytes present in tumour environment) support tumour progression [Schwartz et al, 2014].
· Tsoli M, Schweiger M, Vanniasinghe AS, Painter A, Zechner R, Clarke S, Robertson G. Depletion of white adipose tissue in cancer cachexia syndrome is associated with inflammatory signaling and disrupted circadian regulation. PLoS One. 2014 Mar 25;9(3):e92966. eCollection 2014.
· Schwartz B, Yehuda-Shnaidman E. Putative role of adipose tissue in growth and metabolism of colon cancer cells. Front Oncol. 2014 Jun 26;4:164. eCollection 2014.
They need to link the observation to adipose tissue inflammation in colorectal cancer, or in other types of cancer. What are the findings from cancer patients?
It is a good question and we inserted a paragraph about this topic.
In patients with cancer, there is strong evidence of a linkage between fat mass and colon cancer. However, the molecular mechanisms that govern this linkage are not fully understood. Adipose tissue is probably a trigger and sustains tumorigenesis by producing pro-inflammatory cytokines, growth factors, metalloproteinases and proangiogenic factors, as demonstrated by our findings of increased VEGF [ Ulrich et al 2018, Ulrich et al 2018]. Together these factors induce a higher risk of colon cancer in humans [Ulrich et al 2018]. Recent evidence suggests that IL-32α mRNA and protein levels are upregulated in the adipose tissue of patients with colon cancer. Circulating levels are also increased. The coculture between adipose tissue of obese patients and immortalised colon cancer humans cells showed increased expression of IL-32α in medium and greater proliferation of tumour cells [ Catalán et al. 2017].
However, this association between adipose tissue inflammation and tumorigenesis is found in other types of cancer. WAT inflammation is associated with increased tumour thickness and vascular invasion in patients with early stage oral tongue carcinoma [(Iyengar, 2016].
· Ulrich CM, Himbert C, Holowatyj AN, Hursting SD. Energy balance and gastrointestinal cancer: risk, interventions, outcomes and mechanisms. Nat Rev Gastroenterol Hepatol. 2018 Nov;15(11):683-698. doi: 10.1038/s41575-018-0053-2. Review. PubMed PMID: 30158569.
· Catalán V, Gómez-Ambrosi J, Rodríguez A, Ramírez B, Ortega VA, Hernández-Lizoain JL, Baixauli J, Becerril S, Rotellar F, Valentí V, Moncada R, Silva C, Salvador J, Frühbeck G. IL-32α-induced inflammation constitutes a link between obesity and colon cancer. Oncoimmunology. 2017 May 16;6(7):e1328338.
· Catalán V, Gómez-Ambrosi J, Rodríguez A, Ramírez B, Izaguirre M, Hernández-Lizoain JL, Baixauli J, Martí P, Valentí V, Moncada R, Silva C, Salvador J, Frühbeck G. Increased Obesity-Associated Circulating Levels of the Extracellular Matrix Proteins Osteopontin, Chitinase-3 Like-1 and Tenascin C Are Associated with Colon Cancer. PLoS One. 2016 Sep 9;11(9):e0162189. eCollection 2016.
· Iyengar NM, Ghossein RA, Morris LG, Zhou XK, Kochhar A, Morris PG, Pfister DG, Patel SG, Boyle JO, Hudis CA, Dannenberg AJ. White adipose tissue inflammation and cancer-specific survival in patients with squamous cell carcinoma of the oral tongue. Cancer. 2016 Dec 15;122(24):3794-3802.

Reviewer 2 Report
In this study, authors investigated the effect of butyrate on colorectal cancer by supplementing fructooligosaccharide (FOS) and tributyrin (BUT) in AOM and DSS-induced colon cancer mice model. They tried to characterize the effect of these diet on tumorigenesis by mediating WAT. It is interesting study and give useful information regarding butyrate in colorectal cancer. However, generally, all the figures are so small to read and understand. English needs to be proofread. If tributyrin and FOS possess calories, different calorie intake between control and either tributyrin and FOS group. Also, tumor multiplicity, size data needs to be shown.
Need English proofreading.
Need to use different alphabet for showing significance difference between groups in one-way ANOVA not *, **.
Please make readable size of figures. All figures should be enlarged.
In Table 1, 100 g Tributyrin diet and 60g of FOS was added in each group. How about calories of Tributyrin and FOS per g? Please show total calorie in each diet? Is this effect from calorie intake difference among the groups?
Why authors use mean ± SD or SEM not SD alone or SEM alone? Is there any reason for this?
Why animal numbers of each group is 9-10 or 5-6 ? range not exact number?
Please show tumor multiplicity and size.
Since body weight is lower in BUT group than other groups. Is it possible this low body weight can be cause of most of the results?
Since macrophage types (M1, M2) can affect cancer development and different inflammatory cytokines are affected these supplementations. It is good to show the macrophage types change (IHC) in the colon.
In this study, brown adipose tissues were not much affected compared with the cancer group. How can you explain this?
Author Response
To Reviewer 2
Thank you for your appointments, they are very interesting and improved our article. We will answer point by point.
Need to use different alphabet for showing significance difference between groups in one-way ANOVA not *, **.
We changed to letters: “Values of p < 0.05 (a), p < 0.01 (b), p < 0.001 (c), and p < 0.0001 (d) were considered statistically significant.” This information is in the text.
Please make readable size of figures. All figures should be enlarged.
Now the figures are enlarged, we hope are better to read all.
In Table 1, 100 g Tributyrin diet and 60g of FOS was added in each group. How about calories of Tributyrin and FOS per g? Please show total calorie in each diet? Is this effect from calorie intake difference among the groups?
Tributyrin has 6 Kcal/g and FOS don’t have any calories (0 Kcal/g). We add this information in the text and in the diet table. Mice eat about 2.5 grams/day of this diet, by this way, we do not find any statistical differences between tributyrin supplemented group and the other diets.
Why authors use mean ± SD or SEM not SD alone or SEM alone? Is there any reason for this?
We changed all the graphics and put SEM in all. Thanks for this observation.
Why animal numbers of each group is 9-10 or 5-6? range not exact number?
The group butyrate supplemented is the unique with reduction of survival percent as shown in the figure 2D, we wrote this analysis in the results (second paragraph) to be clear that butyrate worsened the survival. By this way, we performed the experiments in the mice that survived all treatment.
Exactly 5 animals survived and 1 of these mice had enough adipose tissue to perform the analysis, for this reason. For example, adipose tissue analysis (TNF, IL-6 and IL-10) in butyrate group had just 4 analysis in the maximum.
Besides, it is important to clarify that each point of the graphics is the representation of one sample and the reader can observe the specific N of each analysis.
The number are not exactly because we removed outlier not significant, before the anova-one way was applied.
Please show tumor multiplicity and size:
This is a good point, but this carcinogenesis protocol is well established in the literature, so we didn’t perform this analysis, however we have images of tumor presence. 100% of the mice had tumor presence in the colon independent of the diet. The image below show the presence of the tumor in large intestine of the mice during the preparation for histological procedures. The yellow arrow is indicating the tumor.
Since body weight is lower in BUT group than other groups. Is it possible this low body weight can be cause of most of the results?
It is a great point. The adipose tissue depletion and skeletal muscle atrophy are symptoms of cachexia, and these finds are associated with a reduction of cancer patient’s survival. In our study, butyrate worsened the reduction of adipose depots and the gastrocnemius weight.
Since macrophage types (M1, M2) can affect cancer development and different inflammatory cytokines are affected these supplementations. It is good to show the macrophage types change (IHC) in the colon.
It is a great point. We used two different antibody to marker the F4/80 but the background was very high. So, after several attempts and unsuccessful with the different protocols, we gave up. Moreover, we observed constantly in our lab that the surface markers for determination of M1 or M2 macrophages is not correlated with the types of cytokines produced and secreted by adipose tissue.
In this study, brown adipose tissues were not much affected compared with the cancer group. How can you explain this?
In cancer, the white adipose tissue can transforming in beige adipose tissue, this transformation is known as browning, which adipocytes express UCP-1 (uncounpling protein 1) in mitochondria that is a marker of brown adipose tissue.
Beige adipocytes contribute to elevation of energy expenditure leading to negative energy balance (Argilés et al, 2018). In our study, cancer lead to reduction of this organ mass probably because the white adipose tissue were doing the functions of brown adipose tissue, like thermogenesis.
Argilés, J. M., Stemmler, B., López-Soriano, F. J., & Busquets, S. (2018). Inter-tissue communication in cancer cachexia. Nature Reviews Endocrinology. doi:10.1038/s41574-018-0123-0
PS: All the changes in the text are marked. We sent the article for Enago to correct English errors.

Round 2
Reviewer 1 Report
The response from the authors are satisfactory.
Author Response
We sent the manuscript to a company to correct English errors.
Thanks for your appointments, they were interesting and improved our article.
Reviewer 2 Report
Most of questions are answered. But several questions are needed to be answered to improve this manuscript.
Author said that Tributyrin diet has 6 kcal/g and total energy is 600 kcal/kg more than other two diets. However, Tributyrin group animals showed less body weight and adpose tissue weignt. Please explain and discuss the mechanism of this and how affect this in carcinogenesis in the Discussion section.
Author said they removed the outlier before ANOVA, they need to clearly state how many outliers were removed. How they removed? It should be clearly presented in the mansucript. For example, Fig. 2F, the points in FOS group showed 7 points, but they said 9-10 animals in the figure legend. Fig 8 D, only 4 points are shown in FOS group, but they said 9-10 points. Fig 9D showed only 5 points in FOS group, but they said 9-10 points. It is not acceptable to have different animal number in each figure.
Regarding the anser about the lower body weight in BUT group due to low body weight. This answer should be presented in the Discussion since it is important discussion.
Author Response
We would like to thank reviewer for your appointments and the constructive comments that helped us to improve the manuscript. We have explained below point-by-point responses.
Author said that Tributyrin diet has 6 kcal/g and total energy is 600 kcal/kg more than other two diets. However, Tributyrin group animals showed less body weight and adipose tissue weight. Please explain and discuss the mechanism of this and how affect this in carcinogenesis in the Discussion section
Thank you for this appointment, we believe that it is a great point in the discussion. We insert the paragraph with discussion section (lines 395-406):
“It is interesting that butyrate-supplemented group had a consumption additional calorie, about 15kcal/day, by tributyrin added in chow diet. However, this group showed severe reduction on body weight and adipose tissue mass. The literature showed butyrate is absorbed by rectal colon cells in special by the monocarboxylate transporters (MCT)-1 (Inagaki et al. 2018). The butyrate absorption could be inhibited in tumor colonic cells, whereas MCTs are associated with efflux and influx of lactate that is increased in colon cancer cells (Graboń et al 2016). Moreover, patients with intestinal bowel diseases display downregulation of bile acid uptake transporter (Jahnel et al 2014), increased intestinal permeability and disruption of intestinal barrier (Gassler et al 2001), demonstrating bowel inflammation can alters digestion and nutrients uptake.”
Gassler N, Rohr C, Schneider A, Kartenbeck J, Bach A, Obermüller N, Otto HF, Autschbach F (2001) Inflammatory bowel disease is associated with changes of enterocytic junctions. Am J Physiol 281:G216–G228
Jahnel J, Fickert P, Hauer AC, Högenauer C, Avian A, Trauner M.Inflammatory bowel disease alters intestinal bile acid transporter expression. Drug Metab Dispos. 2014 Sep;42(9):1423-31.
Graboń W, Otto-Ślusarczyk D, Chrzanowska A, Mielczarek-Puta M, Joniec-MaciejakI, Słabik K, Barańczyk-Kuźma A. Lactate Formation in Primary and Metastatic Colon Cancer Cells at Hypoxia and Normoxia. Cell Biochem Funct. 2016 Oct;34(7):483-490.
Inagaki A, Hayashi M, Andharia N, Matsuda H. Involvement of butyrate in electrogenic K(+) secretion in rat rectal colon. Pflugers Arch. 2018 Sep 25. [Epub ahead of print]
Author said they removed the outlier before ANOVA, they need to clearly state how many outliers were removed. How they removed? It should be clearly presented in the mansucript. For example, Fig. 2F, the points in FOS group showed 7 points, but they said 9-10 animals in the figure legend. Fig 8 D, only 4 points are shown in FOS group, but they said 9-10 points. Fig 9D showed only 5 points in FOS group, but they said 9-10 points. It is not acceptable to have different animal number in each figure
After the English correction, we typed wrong numbers, in the first version of the manuscript are correct. For this reason, we return the subtitles to the first version that is correct in the figures 6, 7, 8, 9, 10 and 11.
The analysis in colon were made in 5 mice of each group, this tissue is very small, and we have to do analysis with reduced N.
N is different accordingly to the adipose tissue depot, because carcinogenesis experimental protocol damage the mass of these organs, becoming difficult to obtain cytokines results from all samples. The reduced number N are principally in the mesenteric WAT and in butyrate group, butyrate group survived just 5 mice until the end of treatment and mesenteric WAT is difficult to analyze because they are difficult to digest in the buffer.
The outliers were detect by Grubbs’ test and we removed 1 outlier of each group in lipid profile graphs (figure 4), outliers sample are the same mice. In the figures 6, 10 and 11 we also removed 1 outlier in the mesenteric WAT analysis. We wrote this in the line 193.
In figure 2F, we analyze the water consumption by a mean of all mice/2 weeks, by this way there were 7 points besides the N is higher.
About English corrections:
We sent the manuscript to a company to correct English errors, the certificate was attached in our first answers (round 1).
Thanks for your appointments.

Round 3
Reviewer 2 Report
Tha manuscript is well improved as the reviewer suggested.